# Acoustic Optical Fiber Sensor Based on Graphene Oxide Membrane

**DOI:** 10.3390/s21072336

**Published:** 2021-03-27

**Authors:** Catarina S. Monteiro, Maria Raposo, Paulo A. Ribeiro, Susana O. Silva, Orlando Frazão

**Affiliations:** 1Institute for Systems and Computer Engineering, Technology and Science (INESC TEC) and Department of Physics and Astronomy, Faculty of Sciences, University of Porto, Rua do Campo Alegre 687, 4169-007 Porto, Portugal; susana.o.silva@inesctec.pt; 2Faculty of Engineering, University of Porto, R. Dr. Roberto Frias, 4200-465 Porto, Portugal; 3Centro de Física e Investigação Tecnológica (CEFITEC), Departamento de Física, Faculdade de Ciências e Tecnologia, Universidade Nova de Lisboa, 2829-516 Caparica, Portugal; mfr@fct.unl.pt (M.R.); pfr@fct.unl.pt (P.A.R.)

**Keywords:** Fabry–Pérot interferometer, fiber optic sensor, acoustic sensor

## Abstract

A Fabry–Pérot acoustic sensor based on a graphene oxide membrane was developed with the aim to achieve a faster and simpler fabrication procedure when compared to similar graphene-based acoustic sensors. In addition, the proposed sensor was fabricated using methods that reduce chemical hazards and environmental impacts. The developed sensor, with an optical cavity of around 246 µm, showed a constant reflected signal amplitude of 6.8 ± 0.1 dB for 100 nm wavelength range. The sensor attained a wideband operation range between 20 and 100 kHz, with a maximum signal-to-noise ratio (SNR) of 32.7 dB at 25 kHz. The stability and sensitivity to temperatures up to 90 °C was also studied. Moreover, the proposed sensor offers the possibility to be applied as a wideband microphone or to be applied in more complex systems for structural analysis or imaging.

## 1. Introduction

Fiber optic acoustic sensors have been extensively investigated for different applications due to the small size, immunity to electromagnetic interference, and resistance to harsh environments provided by optical fiber. Acoustic sensors are an essential element with a wide range of applications in geophysics for seismic event detection [1,2], or for petroleum exploration [3] and in pipeline leakage [4]. This sensors also play an important role in underwater acoustic sensing [5], with applications in the military field [6], for example, in biomedical applications [7,8,9], and in structural health monitoring [10,11,12].

Optical fiber sensors for acoustic waves detection is based in different mechanisms including photoacoustics [13], with special interest in the biomedical field [14,15], interferometry [16,17], among other [18,19,20]. In particular, interferometry-based sensors have found many research interests due to the compact size and simple fabrication. Most interferometric acoustic sensors are composed by a Fabry–Pérot cavity with a diaphragm or membrane acting as a mirror, capable of deflecting when exposed to pressure variations. Mechanical, optical, and chemical properties of the membrane, along with its geometry, are of utmost importance in the sensor design as it determines the sensitivity, working range, and stability of the sensor, among other sensor characteristics. Different membrane materials have been proposed in literature for the fabrication of Fabry–Pérot based sensors, like fused silica [21,22], metals [23,24], or polymers [25]. Recently, there has been much research on graphene and graphene-based materials due to the compelling material properties. Several works based on Fabry–Pérot interferometers with graphene and graphene-based materials have already been developed for different sensing applications [26,27,28]. Among these, acoustic fiber sensors based on graphene diaphragms have been proposed recently in literature [29,30,31]. Even though pristine graphene offers better mechanical properties when compared to graphene-based materials, such as graphene oxide (GO) or reduced graphene oxide (r-GO), the diaphragm fabrication relies on expensive and time-consuming techniques, such as chemical vapor deposition (CVD) and chemical etching. To overcome this limitation, Yu Wu et al. [32] proposed a microphone based on a GO membrane with controlled thickness around hundreds of nanometers. The fabrication technique is initiated by dissolving GO powder in deionized water, at a controlled concentration. The concentration of the GO solution can be used to control the membrane. Afterwards, the GO solution is placed on a copper foil and heated to remove water and form a membrane. The copper substrate is then removed by chemical etching. The GO membrane is finally placed afloat in a deionized water container and then attached to a glass tube via Van der Waals force. The proposed sensor attained a minimum detectable pressure of 10.2 µPa/Hz^1/2^, and a flat response for frequencies between 100 Hz and 20 kHz. Although the fabrication method of the proposed sensor does not rely on CVD, it still needs chemical etching, and the attained device frequency range is modest from what one can expect for such a membrane thicknesses.

In this work, an acoustic sensor based on a graphene oxide membrane is presented. The proposed sensor is fabricated using the method similar to the recently proposed by Chen Li et al. [27]. Contrary to acoustic sensors based on graphene oxide membranes presented in literature, in this work the membrane is fabricated using a simpler procedure that does not rely on tricky chemical etching, and, consequently, reduces chemical hazards. The diaphragm is dip coated to a 246 µm length silica capillary. The frequency response was investigated in the 20 to 100 kHz frequency range, where a maximum signal-to-noise ratio of 32.7 dB at 25 kHz was achieved. The influence of temperature on the performance of the sensor and in membrane stability was also studied for temperatures between 16 °C and 90 °C.

## 2. Materials and Methods

### 2.1. Sensor Fabrication

The sensor assembly procedure can be divided in two processes: the air cavity assembly and the deposition of the GO membrane. The air cavity was achieved by fusion splicing a single mode fiber (SMF) to a hollow silica capillary. The silica capillary presented an inner diameter of around 75 µm and an outer diameter of 125 µm, equal diameter as the SMF. The fusion splicing process was achieved using a fusion splicer machine (Type-71C, Sumitomo Electric, Osaka, Japan) in a manual setting. The fusion process was performed centered at the SMF, using a lower power electric arc. This process avoids the collapse of the capillary in order to minimize losses at the interface [33]. Afterwards, the capillary was cleaved to the desired length using a fiber cleaver. The cleaving process was controlled by a stereoscopic to analyze capillary length as well as cleave quality.

The deposition of the GO membrane, followed a similar method presented by Cheng Li et al. [27]. This process was carried out by dip coating the capillary free end face in a water dispersion of GO with a concentration of 4 mg/mL. The GO was acquired from Sigma Aldrich and diluted in ultra-pure water (Milli-Q water, Merck KGaA, Darmstadt, Germany). In order to achieve a better control and reproducibility of the dip coating process, the sensor was mounted on a translation stage with controllable velocity. Afterwards, the fiber was placed in an oven at 60 °C for one hour to allow water evaporation, forming the GO diaphragm at the tip of the capillary through Van der Walls interactions [27,34]. By forming the GO membrane directly on the sensor, chemical etching and transferring processes are eliminated. The attained sensing device structure is presented in Figure 1a.

A scanning electron microscope (SEM) image of a GO membrane is also presented in Figure 2a. The attained membrane is mainly homogeneous with a central region with higher density. The radius of the membrane is determined by the internal radius of the silica capillary, which is 75 µm, and the thickness of the membrane can be estimated using SEM images of ruptured membranes as presented in Figure 2b. For this, several membranes were fabricated and ruptured using an ASE (amplified spontaneous emission) light source with 100 mW peak power. An approximated value of 40 nm was attained, less than half of the value presented by Yu Wu et al. [32].

### 2.2. Working Principle

#### 2.2.1. Fabry–Pérot Interferometer

The sensor composed by a GO membrane deposited on a silica capillary that acts as a spacer between the membrane and the optical fiber. The deflections suffered by the membrane due to pressure variations lead to a change of the sensor optical cavity length. This, in turn, translates into a wavelength shift of the reflected spectrum.

Considering the low reflectivity of the graphene oxide diaphragm and in the interface between silica and air, as well as the low diaphragm thickness, the sensor can be regarded as a two-mirror Fabry–Pérot. In this structure, the total reflected light intensity is given by:(1)$$I_{r}\left(\lambda \right)=I_{1}+I_{2}-2\sqrt{I_{1}I_{2}}\cos\left(\frac{4\pi \mathrm{nL}}{\lambda }\right),$$
where $I_{1}$ and $I_{2}$ are the intensity of the light reflections at silica/air and in air/GO interface correspondingly, L is the cavity length, n is the cavity refractive index and λ is the light wavelength.

#### 2.2.2. Acoustic Transduction Mechanism

Acoustic waves generate a variation of pressure near the sensor leading to small diaphragm deflections proportional to the applied acoustic pressure. In turn, the diaphragm deflection induces a cavity length variation that translates into a wavelength shift of the reflected spectrum.

Considering the geometry of the sensor, and assuming a flat and uniform diaphragm thickness, an homogeneous diaphragm material and small deflections, the displacement of the diaphragm at a distance r from the center of the diaphragm caused by an applied pressure with modulus equal to P can be calculated by [35]:(2)$$d\left(r\right)=\frac{3\left(1-\mu^{2}\right)P}{16Eh^{3}}{\left(R^{2}-r^{2}\right)}^{2},$$
where *µ* is the Poisson′s ratio, E is the Young′s modulus, h is the thickness, and R is the diaphragm radius. The maximum deflection, that occurs at the center of the diaphragm (r=0) is linearly proportional to the applied pressure, for frequencies far below the first resonance frequency, which can be obtained from [35]:(3)$$df_{mn}=\frac{k_{mn}^{2}h}{4\pi R^{2}}\sqrt{\frac{E}{3\rho \left(1-\mu^{2}\right)}},$$
where ρ is the mass density of the diaphragm, and $k_{mn}$ is a constant coefficient dependent on the vibration mode. For the first resonant mode, this parameter is given by $k_{11}=3.196$.

In order to determine the operation range of the sensor, the first natural frequency was calculated. For GO, the Young′s modulus is set around 78 GPa [32], the Poisson′s ratio is 0.165, and the mass density $2.2\times 10^{3}\;\mathrm{kg}/m^{3}$ [31]. Considering a membrane diameter of 75 µm with 40 nm thickness, the first resonant frequency is approximately 288 kHz. Ideally, the resonant frequency should be at least three times higher than the input frequency [15], meaning that the maximum operation frequency should be around 96 kHz.

## 3. Results and Discussion

### 3.1. Fabry–Pérot Interferometer

The interrogation system used for optical characterization of the optical cavity was composed by a broadband source, an optical circulator, and an optical spectrum analyzer (AQ6370C, Yokogawa Electric Corporation, Tokyo, Japan) with a resolution of 0.5 nm, as presented in Figure 3.

The reflected spectrum, shown in Figure 4, presented constant amplitude of 6.8± 0.1 dB and a free spectral range of 2.5 nm, corresponding to an optical cavity length of approximately 246 μm. The value obtained by the reflected spectrum is in accordance with the measured value through the microscope image.

### 3.2. Frequency Response

The sensor frequency response was studied using an interrogation system composed by a tunable laser source (TL200C, Thorlabs, New Jersey, United States), an optical circulator, an InGaAs photodetector (PDA10CS-EC, Thorlabs, New Jersey, United States) with variable gain and an oscilloscope, demonstrated in Figure 5. A wideband electric microphone was also used for comparison to the optical signal. The laser source was tuned at 1552.5 nm, a linear region of the reflected spectrum as demonstrated in Figure 4, represented by the dashed line. This was carried out to ensure a linear response of the optical signal to the applied pressure. The acoustic waves with fixed amplitude and variable frequency were produced by a function generator connected to a piezoelectric actuator (P286, Physik Instrumente, Karlsruhe). The frequency of the acoustic waves was varied between 20 and 100 kHz, a frequency region sufficiently far from the resonant frequency of the structure. The pressure sound level was maintained constant at a value of 2.4 Pa, value calculated at 20 kHz, assuming a constant sound level sensitivity of 10 mV/Pa (−40 dBV) for the electric microphone.

The signal from the optical fiber sensor was referenced to the electric signal from the microphone, as presented in Figure 6, where the temporal signal from the two microphones (electrical and optical) is demonstrated for a frequency of 25 kHz. The signal from the electric microphone also served to ensure that the input acoustic pressure was constant.

The Fast Fourier transform (FFT) of the output signal allows to determine the readout frequency as well as the signal to noise (SNR) of the optical sensor. The FFT for applied frequencies between 20 and 100 kHz is presented in Figure 7a. The SNR was calculated assuming a constant noise level for all frequencies. The SNR values varied between a minimum of 14.3 dB, for a frequency equal to 60 kHz, and a maximum of 32.7 dB, for a frequency of 25 kHz. The SNR values for the studied frequencies is presented in Figure 7b.

### 3.3. Directional Response

The directional response of the sensor was studied by varying the angle between the sensor and the piezoelectric actuator, maintaining frequency and amplitude constant. The directional response, presented in Figure 8, attained a maximum SNR value of 33.4 dB for an angle of around −10° between the sound source and the sensor. This may be due to small misalignment of the sensor regarding the sound source position. For angles of magnitudes between 20° and 90° the SNR decreases monotonically to a minimum of 12 dB at 80° angle.

### 3.4. Dynamic Range

The dynamic range is, per definition, the difference between the minimum detectable pressure (MDP) level and maximum input acoustic pressure readable by the system without distortion or non-linearities. The output voltage of the sensor, for the different applied acoustic pressure is presented in Figure 9. For the studied acoustic pressure range, the output voltage of the sensor exhibited a linear response to acoustic pressure with an R-square of 0.998. Harmonic distortion was observable for acoustic pressures higher than 1.96 Pa (99.82 dB-SPL), indicating that this is the maximum acoustic input of the sensor. The noise-limited MPD was calculated by applying an acoustic wave with frequency of 25 kHz and acoustic pressure of 256 mPa (82.2 dB-SPL). The SNR in this conditions is 37.65 dB, with a resolution of 50 Hz, resulting in a noise-limited MPD of 478 µPa/$\sqrt{\mathrm{Hz}}$ (27.57 dB-SPL). The dynamic range of this sensor is limited by the MDP (27.57 dB) and the maximum acoustic input (99.82 dB), yielding a dynamic range of 72.25 dB.

### 3.5. Temperature Response

To evaluate temperature influence, the sensor was placed in a low-temperature oven and was heated between room temperature (16.5 °C) and 90 °C. The variation of temperature caused a wavelength shift and a decrease of fringe visibility, as can be seen in Figure 10a. The linear shift caused by temperature increase presented a rate of 80.8 ± 0.1 pm/°C as shown in Figure 10b. The achieved temperature sensitivity is in accordance to literature [26], where a sensitivity of 0.87 nm/°C, corresponding to 87 pm/°C, was attained for low temperature range. Temperature increase also induced a decrease of fringe visibility, monotonically decrescent for temperatures up to 45 °C. For higher temperatures, the visibility is maintained constant with a fringe contrast equal to 4 dB. After cooling to room temperature, fringe contrast does not return to the initial value, indicating a decrease of membrane quality.

A comparison between the sensor presented in this work and similar configuration with graphene or graphene oxide membranes is listed in Table 1. The sensor proposed in this work as a significantly lower diaphragm diameter which leads to a lower acoustic pressure sensitivity. As a result, the achieved MDP level is higher when compared to similar sensing structures at comparable frequency values. However, the MDP value could be significantly increased by increasing the diameter of the diaphragm.

## 4. Conclusions

An optical fiber acoustic sensor based on a graphene oxide diaphragm has been demonstrated. The presented sensor was attained through a very simple fabrication procedure, which do not require harmful chemical products, when compared to similar acoustic sensors with graphene-based diaphragms.

The sensor achieved a wideband frequency region of operation between 20 and 100 kHz, which is fairly good when compared to graphene oxide based acoustic sensors. A maximum SNR value of 32.7 dB was attained at a frequency of 25 kHz. Temperature characterization was also carried out, achieving a linear wavelength shift of 80.8 pm/°C for temperatures between 16 and 90 °C.

In future work, temperature compensation mechanisms can be added to reduce the influence of thermal fluctuations on the sensor response. Furthermore, considering that temperature may change the mechanical properties of the diaphragm, dynamic studies can be carried out to evaluate the acoustic response of the sensor at different temperatures. The proposed sensor offers the possibility to be applied as a microphone, or to be integrated in more complex imaging systems in fault detection systems in gas pipelines or engineered structures, for example.

## Figures and Tables

**Figure 1 sensors-21-02336-f001:**
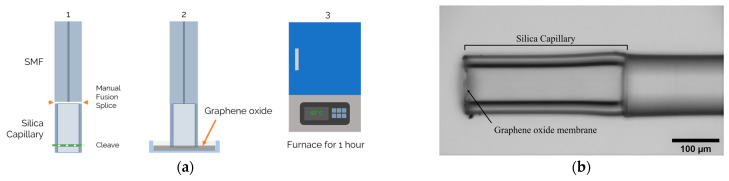
(**a**) Sensor fabrication process: silica capillary fusion spliced to the single mode fiber (SMF) and cleaved to the desired length; graphene oxide (GO) dip coating; and finally, the sensor is placed on a furnace for 1 h at 60 °C; (**b**) microscope photograph of the sensing structure with the GO diaphragm at the tip of the silica capillary.

**Figure 2 sensors-21-02336-f002:**
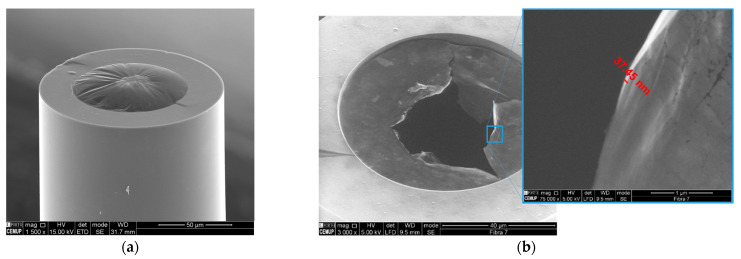
SEM image of a (**a**) sealed GO membrane and (**b**) a ruptured GO membrane and respective thickness evaluation.

**Figure 3 sensors-21-02336-f003:**
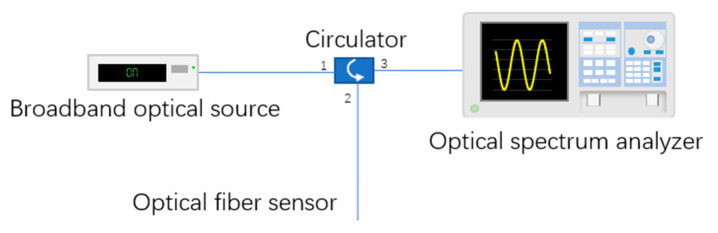
Experimental setup for the Fabry–Pérot interferometer characterization.

**Figure 4 sensors-21-02336-f004:**
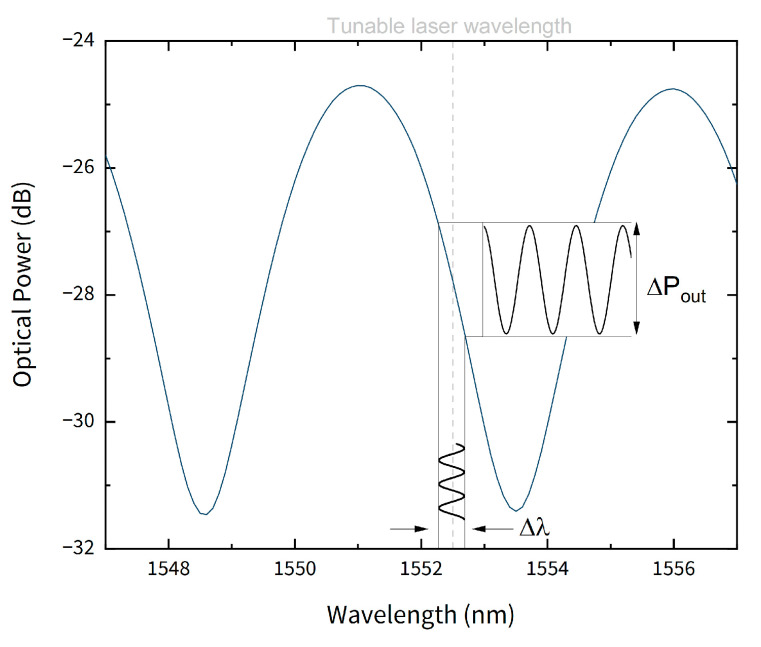
Reflected spectrum of the sensor. The tunable laser wavelength is marked in dashed line.

**Figure 5 sensors-21-02336-f005:**
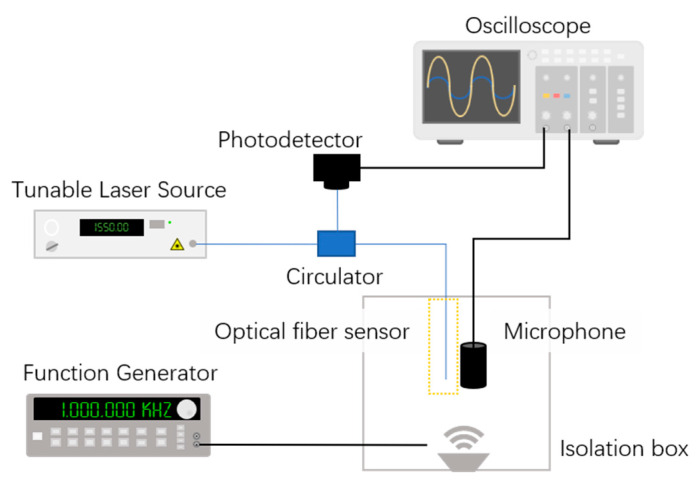
Experimental setup for the acoustic characterization.

**Figure 6 sensors-21-02336-f006:**
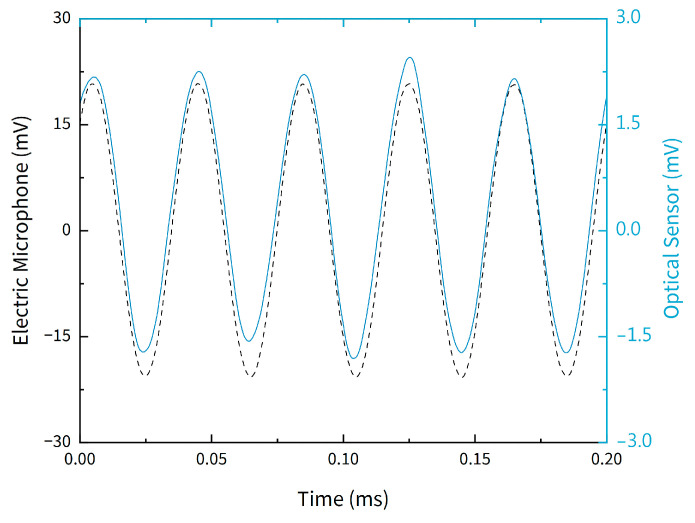
Time domain response of the sensor and the electric microphone to an applied acoustic frequency of 25 kHz.

**Figure 7 sensors-21-02336-f007:**
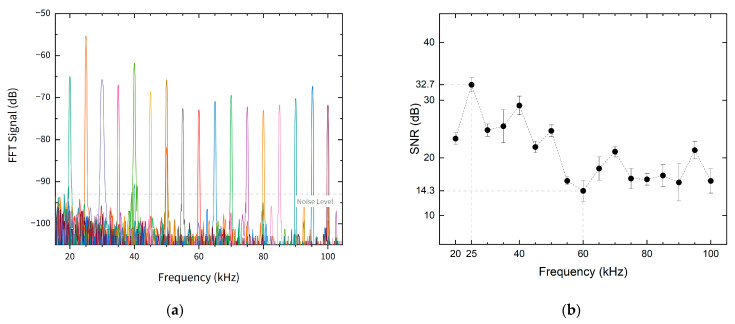
(**a**) Response of the sensor for different acoustic frequency in the frequency domain; (**b**) signal-to-noise ratio (SNR) for the different input acoustic wave frequencies.

**Figure 8 sensors-21-02336-f008:**
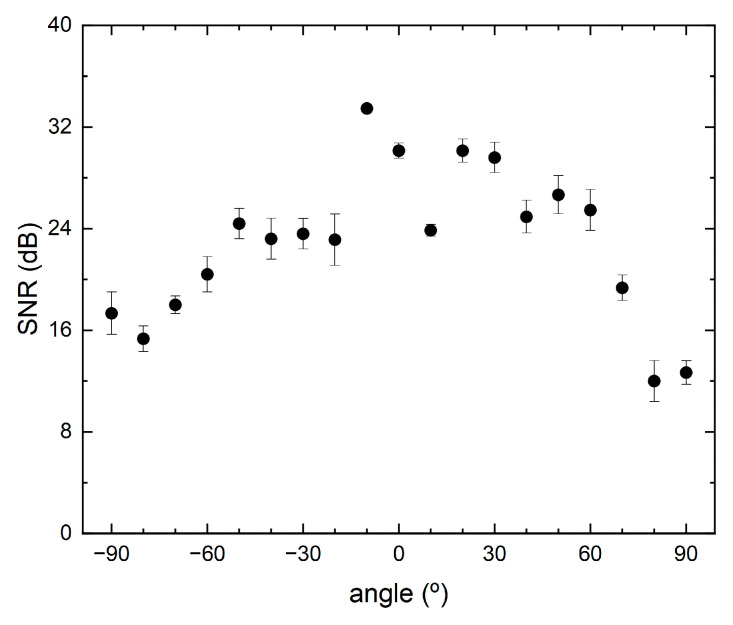
Directional SNR of the sensor for a constant frequency value of 30 kHz.

**Figure 9 sensors-21-02336-f009:**
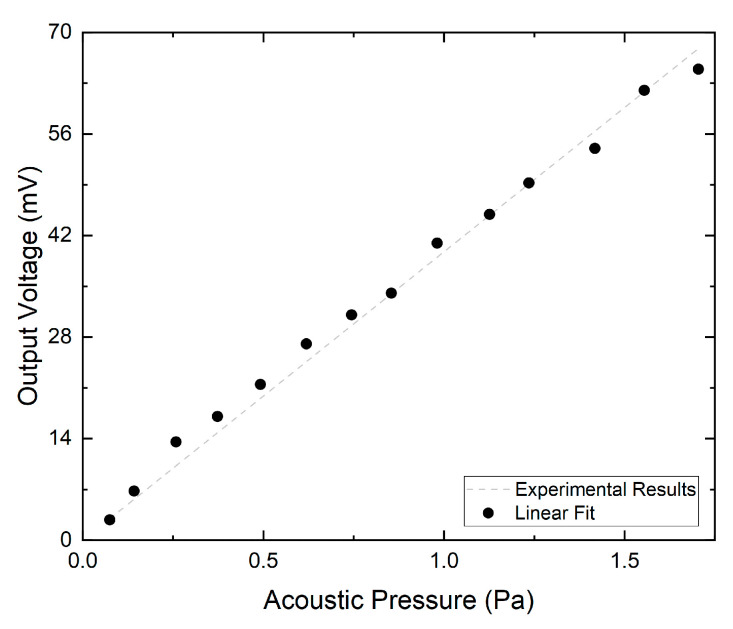
Output voltage of the fiber sensor for the different applied acoustic pressure, at the frequency of 25 kHz.

**Figure 10 sensors-21-02336-f010:**
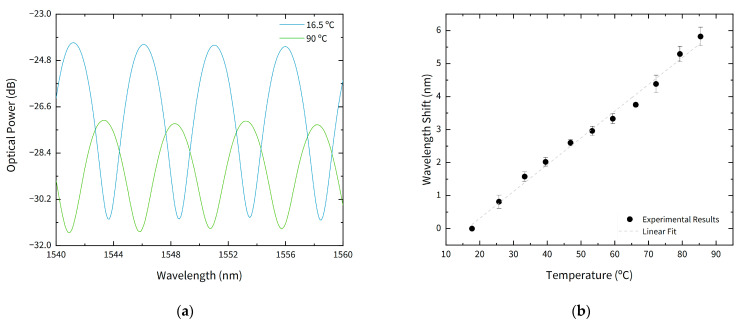
Temperature response of the sensor, for temperatures between 16 °C and 90 °C: (**a**) reflected spectrum for 16.5 °C and 90 °C; (**b**) wavelength shift with temperature.

**Table 1 sensors-21-02336-t001:** Comparison of characteristics and performance between our sensor and similar sensor structures.

Diaphragm Characteristics			MDP	Frequency	Reference
Material	Thickness	Diameter			
Graphene	100 nm	125 µm	59.5 µPa/Hz^1/2^ @10 kHz	0.2 to 22 kHz	[29]
Graphene	10 nm	1 mm	0.77 Pa/Hz^1/2^ @5 Hz 33.97 μPa/Hz^1/2^ @10 kHz	5 Hz to 800 kHz	[30]
Graphene-silver composite	6.36 nm	80 µm	-	0.5 to 30 kHz	[31]
Graphene	4 atomic layers	147 µm	-	0.5 to 20 kHz	
Graphene oxide	100 nm	1.8 mm	1.8 µPa/Hz^1/2^ @20 kHz	0.1 to 20 kHz	[32]
Graphene oxide	~40 nm	75 µm	478 µPa/Hz^1/2^ @25 kHz	20 to 100 kHz	This work

## Data Availability

Data available on request due to restrictions privacy. The data presented in this study are available on request from the corresponding author.

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
