# Peer review of "Acoustic Optical Fiber Sensor Based on Graphene Oxide Membrane"

_sensors, 2021, doi:10.3390/s21072336_

Round 1

Reviewer 1 Report

Authors reported an optical fiber Fabry-Perot intefreometer based acoustic sensor consisting of a graphene oxide membrane. In fact, this sensor structure concept is not new, but a simpler fabrication method for the graphene oxide membrane, that has been developed by other researchers, was used by authors to fabricate the device in current case.  This might be an interesting point for researchers in related areas. However, as for an acoustic sensor, some important performance parameters as well as tesing conditions are missing in the experimental section.  For example, what's the laser wavelength used for F-P operation? What's the sound pressure level used in Fig. 6, Fig. 7 and Fig. 8? How about the sensor response with respect to acoustic pressure? How about the dynamic measurement range for acoustic pressure?Besides the temperature induced change of the reflected spectrum of F-P as shown in Fig. 9, its effect on the final acoustic response of the sensor should aslo be characterized. Moreover, the English writting should be carefully modified for improving readability. 

In summary, this manuscript in current status cannot be considered for publication in Sensors. 

Author Response

Authors reported an optical fiber Fabry-Perot interferometer based acoustic sensor consisting of a graphene oxide membrane. In fact, this sensor structure concept is not new, but a simpler fabrication method for the graphene oxide membrane, that has been developed by other researchers, was used by authors to fabricate the device in current case.  This might be an interesting point for researchers in related areas. However, as for an acoustic sensor, some important performance parameters as well as testing conditions are missing in the experimental section. 

  1. For example, what's the laser wavelength used for F-P operation?

Author’s response: the laser wavelength is represented by the dashed line in Figure 4. The authors agree that this information may not be clear, so additional information was added in page 5, line 168: “The laser source was tuned at 1552.5 nm, a linear region of the reflected spectrum…”.

  1. What's the sound pressure level used in Fig. 6, Fig. 7 and Fig. 8?

Author’s response: The measurements were carried out at constant acoustic pressure. Assuming a constant acoustic pressure sensitivity for the reference microphone of -40 dBV, corresponding to a value of 10 mV/Pa sensitivity, the applied acoustic pressure was around 2.4 Pa. Information of the sound pressure level was added in page 5, line 174: “The pressure sound level was maintained constant at a value of 2.4 Pa, value calculated at 20 kHz and assuming a constant sound level sensitivity of 10 mV/Pa (-40 dB-SPL) for the electric microphone.”

  1. How about the sensor response with respect to acoustic pressure?

Author’s response: The acoustic pressure response was added in section 3.4, page 7. The sensor exhibits a linear response to the applied acoustic pressure in the studied range.

  1. How about the dynamic measurement range for acoustic pressure?

Author’s response: The dynamic range of the sensor was calculated in section 3.4, page 7: “The dynamic range of this sensor is limited by the MDP (27.57 dB) and the maximum acoustic input (99.82 dB), yielding a dynamic range of 72.25 dB.”

  1. Besides the temperature induced change of the reflected spectrum of F-P as shown in Fig. 9, its effect on the final acoustic response of the sensor should also be characterized. Moreover, the English writing should be carefully modified for improving readability. 

Author’s response: The authors agree that the acoustic response of the sensor can be affected by temperature. As demonstrated, temperature induces wavelength shift on the reflected spectrum as well as an irreversible visibility variation. The latter indicates that temperature can deteriorate the membrane, changing its mechanical properties and, therefore, its acoustic response. However, due to the geometry of the oven, it is not possible to induce simultaneous variation of temperature and acoustic pressure. A mention to this characterization was added to the conclusion section, page 8, line 233: “Furthermore, considering that temperature may change the mechanical properties of the diaphragm, dynamic studies can be carried out to evaluate the acoustic response of the sensor at different temperatures.”

Reviewer 2 Report

Dear colleagues, I am pleased to read your communication. It is well structured, contains consistently presented material, confirmed by the results of experiments.
However, a number of questions arose that require clarification.
1. You agree that the main advantage of the sensor is its manufacturing technology, which does not require the use of chemical technologies. In my opinion, this is a very poor argument. Undoubtedly, green technologies are important, but if the resulting sensor has metrological characteristics worse than those of its predecessors, then this solution is not very effective.
2. In this regard, a fairly rich introduction with references to known solutions requires a comparative table on the metrological characteristics of such sensors, indicating the place of your sensor in them. If your sensor is not the best, no big deal, the most important thing is to show that it is not the worst.
3. And, finally, a large volume of the article is devoted to the study of the signal-to-noise ratio. However, the article does not say anything about the nature of the noises. Hence all the information about their change, elimination, etc. becomes meaningless.
I think that these recommendations, if you are positive about them, will help the reader to better understand the level of research you have done.

Author Response

Dear colleagues, I am pleased to read your communication. It is well structured, contains consistently presented material, confirmed by the results of experiments. However, a number of questions arose that require clarification.

  1. You agree that the main advantage of the sensor is its manufacturing technology, which does not require the use of chemical technologies. In my opinion, this is a very poor argument. Undoubtedly, green technologies are important, but if the resulting sensor has metrological characteristics worse than those of its predecessors, then this solution is not very effective.

Author’s response: The reviewer presents a pertinent argument. From the proposed work and the obtained results, it is our conviction that the fiber technology using a graphene oxide membrane has the potential for acoustic sensing. The fabrication process using graphene oxide it’s simple and easy to perform and combined with the unique properties of the proposed material, it opens new opportunities to a wide range of sensing applications.

  1. In this regard, a fairly rich introduction with references to known solutions requires a comparative table on the metrological characteristics of such sensors, indicating the place of your sensor in them. If your sensor is not the best, no big deal, the most important thing is to show that it is not the worst.

Author’s response: we agree with the reviewer’s opinion, and the following comparison table was added in page 9.

  1. And, finally, a large volume of the article is devoted to the study of the signal-to-noise ratio. However, the article does not say anything about the nature of the noises. Hence all the information about their change, elimination, etc. becomes meaningless.
    I think that these recommendations, if you are positive about them, will help the reader to better understand the level of research you have done.

Author’s response: We agree with the reviewer’s opinion, please see, sections 3.2 and 3.3. Also, further information was added with the new section 3.4 (Dynamic range).

Round 2

Reviewer 1 Report

Given the replies provided by authors, the modified manuscript can be accepted for publication.

Reviewer 2 Report

Dear colleagues, thank you for your attentive attitude to my proposals.
Undoubtedly, the article began to look very impressive, and not only due to the technologies used, but also due to your presentation of scientifically grounded, with confirmation of quantitative indicators, sensor characteristics and an indication of its place among similar ones.
I recommend the article for publication.